# The global antigenic diversity of swine influenza A viruses

**Nicola S Lewis**[1]*[†], **Colin A Russell**[2†], **Pinky Langat**[3], **Tavis K Anderson**[4], **Kathryn Berger**[2], **Filip Bielejec**[5], **David F Burke**[1], **Gytis Dudas**[6], **Judith M Fonville**[1], **Ron AM Fouchier**[6], **Paul Kellam**[3], **Bjorn F Koel**[7‡], **Philippe Lemey**[5], **Tung Nguyen**[8], **Bundit Nuansrichy**[9], **JS Malik Peiris**[10], **Takehiko Saito**[11], **Gaelle Simon**[12], **Eugene Skepner**[1], **Nobuhiro Takemae**[11], **ESNIP3 consortium**, **Richard J Webby**[13], **Kristien Van Reeth**[14], **Sharon M Brookes**[15], **Lars Larsen**[16], **Simon J Watson**[3], **Ian H Brown**[15], **Amy L Vincent**[4]

[1]Department of Zoology, University of Cambridge, Cambridge, United Kingdom; [2]Department of Veterinary Medicine, University of Cambridge, Cambridge, United Kingdom; [3]Wellcome Trust Sanger Institute, Hinxton, United Kingdom; [4]Virus and Prion Research Unit, National Animal Disease Center, USDA-ARS, Ames, United States; [5]Clinical and Epidemiological Virology, Department of Microbiology and Immunology, Rega Institute for Medical Research, KU Leuven, Belgium; [6]Institute of Evolutionary Biology, University of Edinburgh, Edinburgh, United Kingdom; [7]Department of Viroscience, Erasmus Medical Center, Rotterdam, Netherlands; [8]Department of Animal Health, National Centre for Veterinary Diagnostics, Hanoi, Vietnam; [9]National Institute of Animal Health, Bangkok, Thailand; [10]School of Public Health, The University of Hong Kong, Hong Kong Special Administrative Region, China; [11]National Institute of Animal Health, Ibaraki, Japan; [12]Swine Virology Immunology Unit, Anses, Ploufragan-Plouzané Laboratory, Ploufragan, France; [13]St Jude Children's Research Hospital, Memphis, United States; [14]Laboratory of Virology, Faculty of Veterinary Medicine, Ghent University, Ghent, Belgium; [15]Animal and Plant Health Agency, Weybridge, United Kingdom; [16]National Veterinary Institute, Technical University of Denmark, Frederiksberg, Denmark

*For correspondence: nsl25@cam.ac.uk

[†]These authors contributed equally to this work

Present address: [‡]Department of Medical Microbiology, Academic Medical Center, University of Amsterdam, Amsterdam, Netherlands

Group author details: ESNIP3 consortium See page 12

Competing interests: The authors declare that no competing interests exist.

**Abstract** Swine influenza presents a substantial disease burden for pig populations worldwide and poses a potential pandemic threat to humans. There is considerable diversity in both H1 and H3 influenza viruses circulating in swine due to the frequent introductions of viruses from humans and birds coupled with geographic segregation of global swine populations. Much of this diversity is characterized genetically but the antigenic diversity of these viruses is poorly understood. Critically, the antigenic diversity shapes the risk profile of swine influenza viruses in terms of their epizootic and pandemic potential. Here, using the most comprehensive set of swine influenza virus antigenic data compiled to date, we quantify the antigenic diversity of swine influenza viruses on a multi-continental scale. The substantial antigenic diversity of recently circulating viruses in different parts of the world adds complexity to the risk profiles for the movement of swine and the potential for swine-derived infections in humans.

**eLife digest** Influenza viruses, commonly called flu, infect millions of people and animals every year and occasionally causes pandemics in humans. The immune system can neutralise flu viruses by recognising the proteins on the virus surface, generically referred to as antigens. These antigens change as flu viruses evolve to escape detection by the immune system. These changes tend to be relatively small such that exposure to one flu virus generates immunity that is still effective against other related flu viruses. However, over time, the accumulation of these small changes can result in larger differences such that prior infections no longer provide protection against the new virus.

Influenza A viruses infect a wide variety of birds and mammals. Viruses can also transmit from one species to another, which may result in the introduction of viruses with antigens that are new to the recipient species and which have the potential to cause substantial outbreaks. Pig flu viruses have long been considered to be a potential risk for human pandemic viruses and were the source of the 2009 pandemic H1N1 virus. Importantly, humans often transmit flu viruses to pigs. Understanding the dynamics and consequences of this two-way transmission is important for designing effective strategies to detect and respond to new strains of flu.

Influenza A viruses of the H1 and H3 subtypes circulate widely in pigs. However, it was poorly understood how closely related swine and human viruses circulating in different regions were to one another and how much the antigens varied between the different viruses.

Lewis, Russell et al. have now analysed the antigenic variation of hundreds of H1 and H3 viruses from pigs on multiple continents. The antigenic diversity of recent swine flu viruses resembles the diversity of H1 and H3 viruses observed in humans over the last 40 years. A key factor driving the diversity of the H1 and H3 viruses in pigs is the frequent introduction of human viruses to pigs. In contrast, only one flu virus from a bird had contributed to the observed antigenic diversity in pigs in a substantial way.

Once in pigs, human-derived flu viruses continue to evolve their antigens. This results in a tremendous diversity of flu viruses that can be transmitted to other pigs and also to humans. These flu viruses could pose a serious risk to public health because they are no longer similar to the current human flu strains. These findings have important implications not only for developing flu vaccines for pigs but also for informing the development of more-effective surveillance and disease-control strategies to prevent the spread of new flu variants.

## Introduction

Swine have been hypothesized to be a mixing vessel for influenza viruses and were the source of the 2009 H1N1 human pandemic virus (*Garten et al., 2009*; *Smith et al., 2009*). The directionality and relative threat of transmission is generally assumed to be from swine into humans with attention only recently turning to introductions in the opposite direction – from humans into swine – and the role that such anthroponotic introductions play in influenza virus evolution (*Nelson et al., 2012*; *Terebuh et al., 2010*). Understanding the dynamics of influenza viruses at the interface between humans and swine is key for designing optimal surveillance and control strategies.

Currently there are three dominant subtypes of influenza A viruses circulating enzootically in swine: H1N1, H1N2, and H3N2. The evolutionary history of these viruses reflects multiple introductions of influenza viruses into swine from other species (*Nelson et al., 2012*; *Vincent et al., 2014*; *Grøntvedt et al., 2013*; *Cappuccio et al., 2011*; *Ottis et al., 1982*; *Pereda, 2010*; *Njabo et al., 2012*; *Karasin et al., 2006*; *Hofshagen, 2009*; *Holyoake et al., 2011*; *Forgie et al., 2011*; *Pensaert et al., 1981*). Much of the work on characterizing swine influenza viruses and their ancestry has been carried out in the U.S and in Europe (*Vincent et al., 2014*; *Marozin et al., 2002*; *Nelson et al., 2014*; *Simon et al., 2014*; *Brown et al., 1993*; *Brown et al., 1997*; *Brown et al., 1998*; *Webby et al., 2000*; *Webby et al., 2004*; *Vincent et al., 2009a*; *Vincent et al., 2009b*; *Kyriakis et al., 2011*; *Watson et al., 2015*) with evidence for numerous introductions of human seasonal H1 and H3 influenza viruses into swine since 1990. This pattern continued with the 2009 human pandemic H1N1 virus (H1N1pdm09) with at least 49 separate human-to-swine transmission events, some of which appear to have become established in pig populations (*Nelson et al., 2012*).

In addition to viruses from humans, avian influenza viruses also occasionally infect swine. Notably, in 1979, epidemics of an avian H1N1 influenza virus lineage were reported in Belgian swine leading to the establishment of an 'avian-like' H1N1 virus lineage in Europe. This Eurasian avian-like 1C lineage (*Zell et al., 2013*) has continued to circulate in swine until the present day, and been introduced into other geographic areas through international movement of swine (*Pensaert et al., 1981*; *Zhu et al., 2013*; *Nelson et al., 2015c*). Unsurprisingly, genetic diversity and multi-species origins of influenza viruses in swine are not restricted solely to pig populations in the U.S. and Europe. China has the largest swine population of any country and shows similar patterns of frequent introductions of both H1 and H3 influenza viruses from both avian and human hosts as well as introductions of swine lineages from other geographic regions (*Chen et al., 2014*; *Vijaykrishna et al., 2011*; *Zhu et al., 2011*).

Once established in swine, influenza viruses of human or avian origin pose a substantial threat to swine populations' health and may spread to other geographically segregated populations (*Nelson et al., 2015c*). Similarly, these viruses also pose a threat for introduction or re-introduction into the human population (*Nelson et al., 2014*), with associated outbreak and pandemic risks. This risk is exemplified by the re-introduction of the H3N2v (variant) virus with surface glycoprotein genes of human origin to over 300 people in 2011–12 from an H3N2 virus that has been endemic in swine in the U.S. since the late 1990s (*Centers for Disease Control and Prevention, 2011*; *Bowman et al., 2014*).

The influenza virus hemagglutinin (HA) surface glycoprotein is the primary target of the humoral immune response, undergoes antigenic drift over time, and is the major antigenic constituent in influenza vaccines used in both humans and livestock. The genetic heterogeneity among the HAs of influenza viruses circulating in swine in certain geographic regions has been relatively well studied and characterized (*Smith et al., 2009*; *Nelson et al., 2012*; *Cappuccio et al., 2011*; *Nelson et al., 2015c*; *Vijaykrishna et al., 2011*; *Kitikoon et al., 2013*; *Anderson et al., 2015*; *Takemae et al., 2008*; *2013*; *Perera et al., 2013*; *Pereda et al., 2011*). However, the relationships between genetic diversity and antigenic diversity among contemporary swine influenza viruses and their antigenic similarity to human seasonal influenza strains are unknown, largely owing to a lack of antigenic data. Such antigenic data are critical in assessing the phenotype of the virus, any potential cross-immunity with viruses circulating in humans and in swine, and the risk of re-emergence from these hosts. These analyses are also critical in the design of antigenically well-matched vaccines.

In order to help fill this knowledge gap, we compiled the largest and most geographically comprehensive antigenic dataset of swine influenza viruses to date. We then used a Bayesian framework (*Bedford et al., 2014*) to quantify the antigenic diversity of influenza viruses circulating in humans and swine thus generating a framework for exploring the antigenic component of risk of introduction of influenza A viruses from swine into humans and to swine in other geographic regions based on the antigenic characteristics of swine influenza viruses circulating in different parts of the world.

## Results

### Dimensionality of influenza virus antigenic evolution in swine

We used hemagglutination inhibition (HI) assays to antigenically characterize a geographically diverse collection of H1 viruses from swine and humans from 1930–2013 (n=194) and H3 viruses from swine and humans from 1968–2013 (n=379) and genetically sequenced all antigenically characterized viruses. Of these, 101 H1 and 73 H3 swine influenza viruses were newly characterized for this study.

In order to accurately quantify the antigenic variation of influenza viruses from swine we first determined the appropriate number of dimensions for accurately representing their antigenic relationships. Previous work on quantifying the antigenic evolution of seasonal influenza viruses showed that the evolution of H3 viruses in humans, as measured in HI assays, could be accurately visualized in two dimensions (2D) using antigenic cartography (*Smith et al., 2004*). In contrast, previous work on swine and equine H3 influenza viruses found that three dimensions (3D) were necessary to accurately visualize antigenic evolution using antigenic cartography (*Lewis et al., 2011*; *2014*; *de Jong et al., 2007*; *Lorusso et al., 2011*). In this study we extensively tested the dimensionality of H1 and

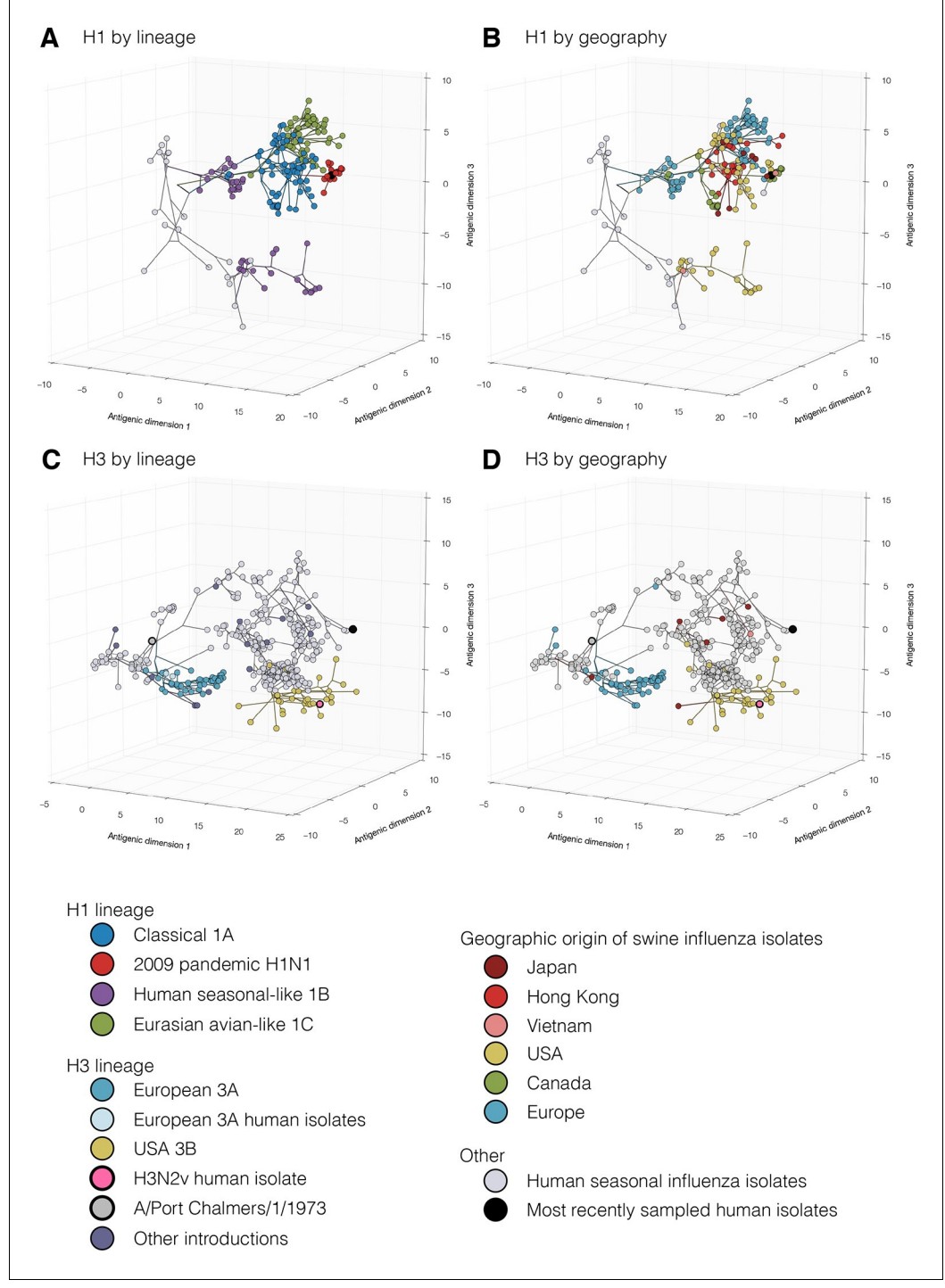

**Figure 1.** Evolutionary relationships of H1 (A, B) and H3 (C, D) influenza viruses circulating in swine and humans inferred by Bayesian Multi-dimensional scaling (BMDS). Each colored ball represents a single virus. Viruses are colored by lineage (**A,C**) and by geography (**B,D**). Lines connecting each virus represent inferred phylogenetic relationships. Distances for antigenic dimensions are measured in antigenic units (AU) and each unit is equivalent to a two-fold dilution in HI assay data. Antigenic distance can be interpreted as a measure of antigenic similarity – viruses close to one another are more antigenically similar than viruses further apart. Interactive visualizations are available at https://phylogeography.github.io/influenzaH1/ and https://phylogeography.github.io/influenzaH3/. Source data and GIF files for rotational views of 3D antigenic maps in *Figure 1* have been deposited in Dryad (*Lewis et al., 2016*).

*Figure 1 continued on next page*

*Figure 1 continued*

The following figure supplements are available for figure 1:

**Figure supplement 1.** *Figure 1A,B* colored by H1 genetic sub lineages in the Bayesian MCC tree.

**Figure supplement 2.** Bayesian H1 MCC tree with taxa labels and posterior support values

**Figure supplement 3.** *Figure 1C,D* colored by H3 genetic sub lineages in the Bayesian MCC tree.

**Figure supplement 4.** Bayesian H3 MCC tree with taxa labels and posterior support values

**Figure supplement 5.** Maximum likelihood phylogenetic trees colored by lineage.

**Figure supplement 6.** Maximum likelihood H1 phylogenetic tree with taxa labels and bootstrap support values

**Figure supplement 7.** Maximum likelihood H3 phylogenetic tree with taxa labels and bootstrap support values.

H3 influenza virus antigenic maps and found that the antigenic variation of both virus subtypes was most accurately represented in three dimensions.

## Antigenic diversity of H1 influenza viruses in swine

To investigate the combined antigenic and genetic evolution of our study viruses we implemented a Bayesian multi-dimensional scaling (BMDS) method for inferring antigenic and genetic relationships (*Bedford et al., 2014*). As of 2013, three major H1 lineages circulated in swine: 1. the classical 1A lineage – descended from the ancestors of the 1918 human influenza pandemic, was first detected in swine in the 1930's, and gave rise to the human H1N1pdm09 viruses and swine H1N1pdm09-like sub-lineage viruses; 2. the human seasonal-like 1B lineage – resulting from multiple introductions from human seasonal H1 viruses; 3. the Eurasian avian-like 1C lineage – arising from an introduction from wild birds into swine and first detected in Europe in the 1980's (*Figure 1A*, *Zell et al., 2013*).

Our combined antigenic and genetic analyses of H1 viruses in swine showed an inconsistent relationship between antigenic clustering and genetic lineage through time with occasional emergence of new antigenic variants. In all geographic regions for which we had antigenic data we found unique patterns of antigenic evolution within individual lineages.

In Asia, we observed co-circulation of classical 1A, human seasonal-like 1B, and Eurasian avian-like 1C lineage viruses. However all three lineages were not detected in any one country. In Hong Kong – where samples were derived at slaughter from swine sourced from China – classical swine 1A lineage including the H1N1pdm09 sub-lineage, and Eurasian avian-like 1C lineage viruses were detected (*Figure 1A*, light red). For the viruses from Hong Kong, the mean pairwise antigenic distance (MPD) between the H1N1pdm09 sub-lineage viruses and the two classical 1A lineage clades was 5.0 and 6.0 AU away (*Figure 1—figure supplement 1*: *Figure 1—figure supplement 2*, *Supplementary file 2*). The Classical swine 1A lineage viruses collected in Japan (*Figure 1A*, dark red) differed from the Hong Kong H1 viruses with a MPD of 5.6 AU. Within the Hong Kong Eurasian avian-like 1C lineage viruses there were two distinct phylogenetic groups – group 1 and group 2 (as previously reported in *Yang et al. (2016)* and *Figure 1—figure supplement 1*, *Supplementary file 2*). The Group 1 and Group 2 viruses were 6.5 AU and 4.4 AU (MPD) away from the H1N1pdm09 sub-lineage viruses. The MPD between all Asian and European Eurasian avian-like 1C lineage strains was 5.2 AU. All of these distances are sufficiently large that it would be surprising if prior infection or un-adjuvanted vaccines would provide any protection among these virus groups.

In Europe, the Eurasian avian-like 1C lineage co-circulated with the H1N1pdm09 sub-lineage of classical 1A viruses and the human seasonal-like 1B lineage. The human seasonal-like 1B lineage viruses were first detected in European swine in the mid-1990's but have a putative human seasonal ancestor from the 1980's. The human seasonal-like 1B viruses differed substantially from the Eurasian avian-like swine 1C viruses and the H1N1pdm09 sub-lineage viruses (MPD: 6.7 AU and 10.4 AU respectively). The H1N1pdm2009 sub-lineage viruses also differed substantially from the human seasonal-like 1B lineage viruses (MPD:10.6 AU).

In Canada, only the classical swine 1A lineages viruses and H1N1pdm09 sub-lineage viruses were detected. Most of the viruses within the classical swine 1A lineage were antigenically similar to each other but differed from H1N1pdm09 sublineages viruses with a MPD of 6.5 AU. However, the alpha sub-lineage of classical 1A lineage viruses (*Figure 1B* green, *Supplementary file 2*) antigenically diverged in Canadian swine to form a separate antigenic group with a MPD of 7.0 AU from the H1N1pdm09 strains and 4.2 AU from the other Canadian classical swine 1A lineage viruses.

In the USA, the classical swine 1A lineage viruses co-circulated with the H1N1pdm09 sub-lineage and the human seasonal-like 1B lineage viruses. In the USA the human seasonal-like 1B lineage viruses evolved two sub-lineages – Delta 1 and Delta 2 (*Figure 1—figure supplement 1*, *Supplementary file 2*) – likely as the result of two separate but nearly contemporaneous introductions into swine first detected in the early 2000's. The Delta 1 and 2 sub-lineage viruses antigenically differed from the H1N1pdm2009 lineage viruses with an MPD of 7.8 AU. Representatives of alpha sub-lineage of the classical swine 1A lineage in the USA were 7.0 AU (MPD) from the H1N1pdm2009 sub-lineage viruses. The beta and gamma sub-lineages of the classical swine 1A lineage were 6.6 and 4.7 AU (MPD) respectively from the H1N1pdm2009 viruses.

To assess the risk of virus introduction associated with moving live swine between continents we compared the MPDs of swine influenza H1 viruses circulating in different parts of the world. The human seasonal-like 1B viruses circulating in European swine were on average 11.7 AU (MPD) from the human seasonal-like 1B viruses in USA swine. The Eurasian avian-like 1C lineage viruses were 6.8 AU (MPD) from the human seasonal-like 1B lineage viruses in the USA.

## Antigenic diversity of H3 influenza viruses in swine

Just as for the H1 viruses, we used BMDS (*Bedford et al., 2014*) to investigate the antigenic and genetic evolution of H3 viruses (*Figure 1C,D*). In European swine there were four H3 introductions included in our antigenic characterization (*Figure 1—figure supplement 3*: *Figure 1—figure supplement 4*). Of these, the European 3A lineage (A/Port Chalmers/1973-like) introduction was the most commonly detected lineage, showing evidence for three sequential antigenic clusters and evolving on a markedly different path from the putative ancestor in humans (*Figure 1C,D* [blue]). A separate introduction from humans into swine of an antigenically distinct 1970's-like human seasonal virus also circulated in European swine in the 1980s and 1990s (*Figure 1—figure supplement 1* [Europe 2 Introduction]). The earliest strain in the European 3A lineage was 6.5 AU from the A/Port Chalmers/1973 reference strain (*Figure 1C* large grey) and the most recent strain was 9.8 AU (*Supplementary file 2*). The distance from our representative of recent human seasonal H3 viruses (A/Victoria/361/2011) to the H3 European virus in swine and was 14.9 AU (MPD).

In the USA, we found evidence for three H3 virus introduction events from humans into swine in the mid-1990's, one of which resulted in the establishment of the 3B lineage (*Figure 1C,D*) which became the most frequently detected lineage in swine through 2013, and gave rise to the H3N2v viruses which infected >300 humans in the US from 2011–2012 (*Centers for Disease Control and Prevention, 2012*). We only antigenically characterizes a single representative of the H3N2v viruses – A/Indiana/8/2011 – and found that it differed from A/Victoria/361/2011 by 6.4 AU (*Figure 1* pink). The MPD between the swine H3 viruses circulating in the USA and in Europe was 12.6 AU.

In Asia we found evidence for multiple introductions of H3 viruses and subsequent antigenic drift from the six representative viruses in our dataset including an introduction likely mediated by translocation of a virus circulating in USA swine to Asian swine (*Nelson et al., 2015d*). Two separate introductions from humans were detected in Japan that resulted in circulation in swine beyond the time of circulation of the ancestral strain in humans. The introductions were ~5.2 AU from A/Victoria/361/2011 and 10.9 AU (MPD) from the H3 strains circulating in European swine. The most recent seasonal human H3 introduction detected in Asia was from Vietnam and this virus was 3.8 AU away from A/Victoria/361/2011 (*Supplementary file 2*).

## Rates of antigenic drift in swine

To investigate the rates of swine influenza virus antigenic drift away from the most recent common ancestor for each genetic lineage in each geographic region we used TimeSlicer available in SPREAD (*Bielejec et al., 2011*). For H1 viruses in swine, the classical 1A lineage evolved at a mean rate of 0.15 AU per year from the earliest antigenically characterized representative in the dataset (A/swine/

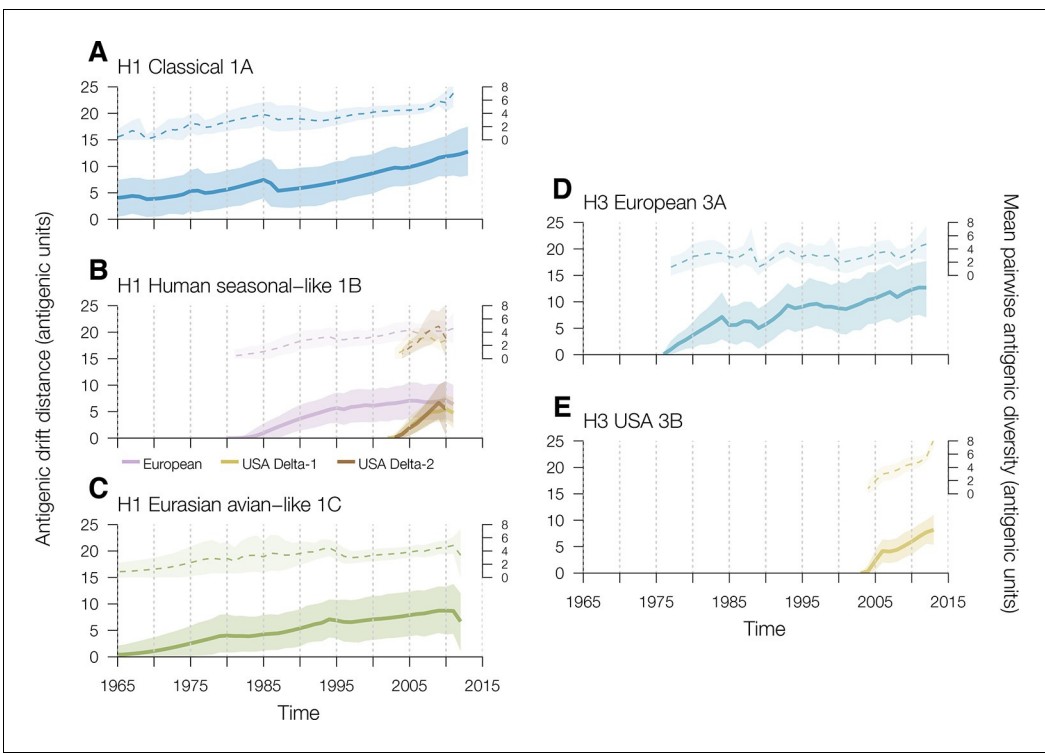

**Figure 2.** Time series of year-to-year rates of antigenic drift distance and antigenic diversity of H1 and H3 viruses in swine by genetic lineage. Solid colored lines represent year-to-year antigenic drift distance, where drift for year *i* is measured as the mean of Euclidean distances among strains in a phylogenetic lineage in year *i* compared to the mean of Euclidean distances among strains of that phylogenetic lineage from the previous year (*i*–1). The dotted line represents antigenic diversity among H1 and H3 strains by lineage through time. For the solid and dotted lines, the shaded region represents the range of the highest posterior density estimates. Multiple introductions which circulate for >5 years of the human seasonal-like swine H1 lineage in European (purple) and USA (gold) swine were calculated separately. Source data for *Figure 2* has been deposited in Dryad (*Lewis et al., 2016*).

Iowa/15/1930) (*Figure 2A*, *Supplementary file 3*). The human seasonal-like 1B viruses introduced into European swine evolved at a mean rate of 0.17 AU per year (*Figure 2B*, *Supplementary file 3*). The introductions of human seasonal H1 viruses into USA swine that gave rise to the Delta-1 and Delta-2 lineages showed slightly different rates of antigenic drift – 0.63 for Delta-1 and 0.85 for Delta-2 – both significantly faster than the human seasonal introduction in European swine (*Figure 2B*, *Supplementary file 3*) but they were only observed over a relatively short time period. The Eurasian avian-like swine 1C lineage evolved at a mean rate of 0.15 AU per year (*Figure 2C*, *Supplementary file 3*). With the exception of the human seasonal-like 1B lineage viruses in the USA, the overall antigenic diversity of H1 viruses in swine remained remarkably constant over the study period (*Figure 2A,B,C*). The antigenic diversity of the Delta-1 and Delta-2 sublineages in the USA began to expand rapidly around 2009 for reasons that remain to be elucidated (*Figure 2B*).

For H3 viruses, we found that the 3B lineages viruses in USA swine evolved antigenically at a faster rate than the H3 viruses introduced into European swine (0.49 AU per year vs 0.28 AU per year [*Figure 2D,E*, *Supplementary file 3*]). Additionally, the antigenic diversity of H3 viruses in USA swine, particularly since 2010, has increased at a faster rate than that of H3 viruses in Europeans swine. Thus sustained circulation of antigenic drift variants within US swine has substantially increased the observed antigenic diversity in this lineage relative to the observed antigenic diversity in other geographic regions.

## Additional antigenic diversity can be inferred from genetic sequence data

In addition to the BMDS analyses of viruses for which we had both antigenic and genetic data, we also performed large-scale phylogenetic analyses of publicly available swine influenza virus sequence. Through these analyses we inferred potentially important gaps in the global antigenic data that suggest our antigenic dataset underrepresents the true global antigenic diversity of swine influenza viruses.

For H1 viruses that circulated in swine as of 2013, we find that the major genetic lineages have been captured in this study (*Figure 1—figure supplement 5*: *Figure 1—figure supplement 6*). However, new genetic evidence recently became available to demonstrate additional human seasonal introductions into swine in previously under-surveilled regions such as Central and South America (*Nelson et al., 2015a*; *Dibárbora et al., 2013*; *Nelson et al., 2015b*) (*Figure 1—figure supplement 5*). Since genetic lineage does not always predict antigenic phenotype, we cannot make any inferences about their antigenicity.

Even in cases where we have antigenically-characterized clade representatives, the long branch lengths associated with some introductions could conceal antigenic drift away from the characterized clade representatives. The markedly different evolutionary patterns seen in the H3 lineages characterized in swine and the inferred antigenic profile of the viruses introduced to swine, suggests that the true diversity of H3 viruses circulating in swine is likely to be substantially greater than the diversity found in our BMDS analyses. In particular, we identified 19 additional examples of introductions of seasonal H3 viruses from humans in 10 countries (*Figure 1—figure supplement 5* – black tips: *Figure 1—figure supplement 7*). Based on the likely time of these introductions from humans into swine we estimate that the introduced viruses represent at least 12 distinct antigenic variants based on the antigenic variation of their human seasonal influenza ancestors. Similar to the H1 genetic lineages described above, these additional H3 lineages also require antigenic testing to interpret the impact of these introductions on the global antigenic diversity of IAV in swine.

## Risk posed by swine influenza viruses to the humans

In addition to the challenge the observed antigenic diversity poses to swine populations, the antigenic diversity of swine influenza viruses also creates substantial risks for human populations. The outbreak and pandemic potential of swine influenza viruses in humans is at least partially determined by human population immunity against swine viruses. This component of risk can be inferred from the antigenic distance of swine influenza viruses to seasonal viruses or vaccine strains in humans to which humans are likely to have immunity. Importantly, individuals are unlikely to have immunity to viruses that only circulated before they were born (*Fonville et al., 2014*).

Since most of the current swine influenza viruses are the product of human seasonal influenza virus introductions into swine, we anticipate at least some cross-protective immunity in the human population that could potentially interfere with re-introduction of these viruses. For example, the H1N1pdm09 viruses circulating in both humans and swine are very similar antigenically and likely induce at least some cross-immunity in both hosts (*Figure 1A,B*). However, for the H1 1C, H3 3A, and H3 3B human seasonal lineages in swine, the risk of re-introduction into the human population increases with the proportion of the human population that were born after the human precursor virus circulated in humans and is amplified by antigenic evolution of these viruses in swine. Thus, in terms of antigenic similarity and likely prior exposure, earlier introduced lineages of human H1 and H3 viruses – particularly those with precursors antigenically similar to the H3 1968 pandemic strain – pose the greatest current risk to humans because of the low or negligible predicted levels of cross immunity in individuals born since the 1970s. In comparison, the H3N2v viruses, which ultimately derive from a human seasonal virus introduction in the 1990s, have predominantly infected people born after the mid-1990s.

## Discussion

In this study we characterized the antigenic diversity H1 and H3 viruses circulating in both swine and humans on a multi-continental scale. The observed diversity was largely driven by frequent introductions of variants from humans into swine and subsequent antigenic evolution within swine combined

with long-term geographic segregation of multiple virus lineages. Much of the antigenic evolution of human-origin viruses in swine was via long and divergent evolutionary paths from those observed in humans. These divergent paths create threats for the re-introduction of viruses into humans as well as the possibility for antigenically novel strains in humans to be introduced into swine. Interestingly, the introduction of viruses from birds to swine has made only one major contribution to the antigenic diversity of influenza viruses in swine. This could be at least partially due to the antigenic similarity of the avian-like and the classical swine 1A lineages found here, although an exhaustive test of antigenic similarity between avian and swine H1 and H3 viruses could not be performed as part of this study and to the best of our knowledge has not been reported elsewhere. Successful introduction of avian influenza viruses into swine will also likely be influenced by other factors including mutations conferring adaption for mammalian transmission (*Herfst et al., 2012*; *Imai et al., 2012*) and other physiological and ecological factors.

In addition to multiple introduction events and subsequent antigenic drift, the patterns of antigenic diversity and heterogeneity among geographic regions are also influenced by the relative geographic isolation of swine populations resulting in region-specific patterns of virus evolution and circulation. The observed patterns of antigenic heterogeneity among geographic regions are also likely influenced by livestock production system differences, particularly regional differences in the movement of swine within and between animal holdings. USA producers commonly move swine among otherwise geographically isolated areas at different ages after weaning, whereas in the EU there is negligible intra- or inter-country movement of animals during a single production cycle. Such livestock movement patterns have been shown to influence the evolution of influenza viruses in swine in the USA, with the Midwest serving as an ecological sink for swine influenza virus with the source of genetic diversity located in the south east and south central states for some HA lineages (*Nelson et al., 2011*).

Vaccination is used extensively as a means to control influenza in swine in the USA and sporadically globally. Control strategies vary by region with some countries having no swine influenza vaccine usage, while others produce autologous (herd specific) vaccine for individual producers, which may be used consecutively with commercially-manufactured multi-strain vaccine when a new outbreak strain requires an update to vaccine composition. Most autologous and commercial products are multivalent due to the diversity of circulating strains. However, there is no formalized system for matching vaccine strains with circulating strains, nor validated protocols for standardization and effective vaccine use. The antigenic diversity quantified here is sufficiently large that it is highly unlikely that one strain per subtype would be efficacious globally, or even within a given region. Multiple within-subtype strains would likely be required to produce a vaccine that afforded adequate protection against most circulating variants using current inactivated vaccine production processes and application protocols.

Interestingly, H3 viruses circulating in European swine showed lower antigenic diversity than other regions. This limited antigenic diversity, combined with the use of oil adjuvanted vaccines, could explain why the prototype A/Port Chalmers/1973 (H3N2)-based vaccine induced protection against contemporary H3N2 swine influenza viruses isolated up to 35 years later (*de Jong et al., 2007*; *Van Reeth and Ma, 2013*; *De Vleeschauwer et al., 2015*). However, movement of swine among geographic regions could lead to the introduction of novel variants into a new sub-population of swine despite current import-export quarantine procedures between some countries. Therefore a more structured approach to vaccine strain selection in swine, where epidemiological, antigenic and genetic data are considered at a global level by the World Organization for Animal Health (OIE), similar to the process undertaken for creating equine influenza vaccine recommendations, could provide an evidence-based rationale for swine vaccine strain selection and improved vaccine efficacy in swine.

Age-related susceptibility within the human population could be one of the limiting factors in onward spread of recent human infections with swine viruses. However there remains a need for focused surveillance in areas with high swine population density and situations where humans and swine have opportunities for close contact to better assess the incidence of cross-species transmission and risk of onward transmission within the human population.

Neutralizing immunity to influenza viruses in both humans and swine is largely directed towards the influenza HA and the antigenic dissimilarity among influenza HAs within and between these two hosts is undoubtedly important for influenza epidemiology. However, the potential for a virus to

emerge and spread widely in human or swine populations is likely to be a multi-genic trait comprising factors beyond just antigenicity (*Trock et al., 2012*). These factors include receptor binding (*Matos-Patrón et al., 2015*; *Elderfield et al., 2014*), adaptations to the matrix, non-structural, and neuraminidase proteins (*Elderfield et al., 2014*) as well as other factors differing between human and swine influenza viruses yet to be identified. There is also a need to better understand the competition dynamics of different influenza virus variants and subtypes in both human and swine populations and how these dynamics affect the potential for invasion by novel viruses.

Quantifying the public health risk of circulating swine influenza viruses in terms of immunological protection could be improved by assessing human population immunity to a selection of the antigenically-diverse swine viruses characterized in this study. A globally coordinated and systematic pipeline of antigenic and genetic analyses of relative antigenic distance between human and swine strains for subsequent testing of highly divergent swine strains against human sera (age-stratified) would add valuable information to the current WHO-led emergent pandemic risk assessment and inform the design of strategic vaccine stockpiles for human populations most at risk of infection with swine influenza virus, whether by age, geographic location or other factors.

## Materials and methods

### Viruses

Using hemagglutination inhibition (HI) assays, we characterized the antigenic properties of swine influenza A H1 and H3 virus strains circulating 1) in Europe, as part of the third program of the European Surveillance Network for Influenza in Pigs (ESNIP3) (www.esnip3.com), and 2) in Asia in collaboration with laboratories in Hong Kong, Japan, Vietnam and Thailand. We also expanded previously published datasets for North America. To this HI assay dataset we added previously published swine and human HI assay data for H1 and H3 influenza viruses (*Lewis et al., 2014*; *de Jong et al., 2007*; *Lorusso et al., 2011*; *Koel et al., 2013*; *Nfon et al., 2011*), resulting in an antigenic dataset consisting of 194 swine and human H1 influenza viruses and 379 swine and human H3N2 influenza viruses.

Viruses were propagated in Madin-Darby canine kidney (MDCK) cells or embryonated fowls' eggs. Harvested cell culture supernatant or allantoic fluid was clarified by centrifugation. For antisera production virus was concentrated by ultracentrifugation over a 20% w/v sucrose cushion. Virus pellets were resuspended overnight at 4°C in sterile phosphate buffered saline at pH 7.4 and stored at -70°C for use in immunizing naïve pigs.

### Swine antisera production

Swine antisera to influenza A viruses were generated by the United States Department of Agriculture National Animal Disease Centre, Ames, Iowa as previously described (*Lorusso et al., 2011*). Additional sera were generated by the French Agency for Food, Environmental and Occupational Health & Safety, Ploufragan, France, and by the Technical University of Denmark, Copenhagen, Denmark. To generate these additional antisera each animal was first inoculated intranasally with live influenza virus (7 to 8 log10 $EID_{50}$ in a volume of 3 ml; 1.5 ml per nostril). Three weeks later equal volumes of the same inoculating virus and Freund's complete adjuvant (total volume 3 ml) or inoculating virus and Montanide ISA206 (Seppic, Givaudan-Lavirotte, France) (total volume 2 ml) were administered by intramuscular/intradermal injection The animals were killed and bled two weeks after the last immunization. For sera raised in the USA, pigs were cared for in compliance with the Institutional Animal Care and Use Committee of the National Animal Disease Centre and for sera raised in the EU, all animal work was done in accordance with the local rules and procedures with ethical permissions granted following the codes of practice for performing scientific studies using animals.

### Virus antigenic characterization

HI assays using swine antisera were performed to compare the antigenic properties of swine and human influenza H1 and H3 viruses. Prior to HI testing, sera used for testing most H1 and all H3 viruses were treated with receptor-destroying enzyme (Sigma-Aldrich, MO, USA) and sera used for testing H1 viruses from the USA were treated with kaolin (Fisher Scientific, Pittsburg, PA, USA). All sera were then heat inactivated at 56°C for 30 min and adsorbed with 50% turkey red blood cells (RBC) to remove nonspecific inhibitors of hemagglutination. HI assays were performed by testing

reference antisera raised to swine influenza viruses against selected H1 and H3 swine and human influenza viruses according to standard techniques. Serial 2-fold dilutions starting at 1:10 were tested for their ability to inhibit the agglutination of 0.5% turkey RBC with four hemagglutinating units of swine and human H1 and H3 viruses.

## Antigenic cartography

To combine HI data generated in different laboratories we used a standard reference panel consisting of viruses and swine antisera raised to selected reference strains and shared among laboratories (*Lewis et al., 2014*; *Lorusso et al., 2011*). Our initial quantitative analyses of the antigenic properties of swine and human influenza H1 and H3 viruses used antigenic cartography methods as previously described for human (H3) and swine influenza A (H3) and (H1) viruses (*Smith et al., 2004*; *Lewis et al., 2014*; *de Jong et al., 2007*; *Lorusso et al., 2011*; *Nfon et al., 2011*).

To identify the most appropriate dimensionality of the antigenic maps for each virus subtype, we made first constructed antigenic maps of H1 and H3 viruses in 1, 2, 3, 4, and 5 dimensions and compared the HI titer differences to the resultant antigenic map distances for each dimension. Increasing from 1 to 2, and from 2 to 3 dimensions, resulted in significant improvement in correlation between HI data and map distances indicating that 3 dimensional (3D) maps provide a more accurate representation of the underlying HI data (*Supplementary file 1*). Further increasing map dimensionality resulted in smaller and statistically insignificant improvements in the quality of the fit of the map to the underlying HI data. Based on these results, we concluded that 3D maps were sufficient for capturing the antigenic variation of both H1 and H3 viruses in our combined swine and human datasets. Next we removed a random 10%, 20%, 30%, 40%, and 50% of the HI data and re-made the antigenic maps to test the robustness of the antigenic maps and the resolution at which they could visualize antigenic distances among viruses. The precision of point positioning in the 3D antigenic maps was 0.68 antigenic units (AU) for H1 and 0.85 AU for H3. One AU is equivalent to a two-fold difference in HI assay titer. Three AUs are considered sufficient antigenic difference to warrant an update of the human influenza vaccine.

## Integrated analysis of antigenic and genetic evolution

For all isolates with both HA sequence data and HI measurements available, we implemented a Bayesian multidimensional scaling (BMDS) cartographic model to jointly infer antigenic and phylogenetic relationships of the viruses as described by *Bedford et al. (2014)*

In brief, time-resolved phylogenies were estimated for a total of 194 H1 and 379 H3 sequences using the Bayesian Markov chain Monte Carlo (MCMC) method implemented in BEAST (*Drummond et al., 2012*). We incorporated the Hasegawa-Kishino-Yano (HKY) model of nucleotide substitution model (*Hasegawa et al., 1985*) with codon partitioning at all three positions, and a Bayesian Skygrid coalescent model (*Gill et al., 2013*). A strict molecular clock and an uncorrelated lognormal relaxed clock were used for H3 and H1, respectively, based on linear regression analyses in Path-o-Gen v1.4 (http://tree.bio.ed.ac.uk/software/pathogen/). Ancestral sequence states were reconstructed using the 'renaissance counting' method (*Lemey et al., 2012*). Two independent MCMC chains were run for 200 million states with sampling every 20,000 states and a burn-in of 20 million states, and then combined and further subsampled every 180,000 states for a total of 2000 trees after assessing convergence in Tracer v1.6 (http://tree.bio.ed.ac.uk/software/tracer/). Along with the matching HI measurements for the virus isolates against post-infection ferret and swine antisera, these subsets of 2000 trees were used as an empirical tree distribution for the subsequent BMDS analysis to simultaneously model antigenic locations and evolutionary history.

The BMDS approach implements a Bayesian analog of antigenic cartographic models from Smith et al. (*Smith et al., 2004*) that arrange virus and serum locations in N-dimensions, such that Euclidean distances between locations are inversely proportional to serological cross-reactivity (*Bedford et al., 2014*). We generated the antigenic maps by imposing a weakly informative prior on expected antigenic locations such that antigenic distance increases with sampling time along one dimension and incorporated genetic data by modelling changes in antigenic phenotype as a diffusion process along the viral phylogeny, i.e. expected virus locations co-vary with genetic relatedness. Virus avidities and serum potencies were also estimated to account for experimental variation in serum and virus reactivity. MCMC was used to sample virus and serum locations in either two or

three antigenic dimensions, as well as virus avidities, serum potencies, antigenic drift rate, MDS precision, virus and serum location precisions, and phylogenetic patterns. MCMC chains were run for 500 million states with sampling every 200,000 states, and checked for convergence by high ESS values in Tracer. Following burn-in of 100 million states, we obtained a total of 2,000 trees from which the maximum clade credibility tree was summarized in TreeAnnotator v1.8.2.

We used TimeSlicer available in SPREAD (*Bielejec et al., 2011*) to quantify the rates of antigenic evolution and levels of standing diversity over the posterior distribution of trees. This was achieved by 'slicing' through the BMDS-derived antigenic map location-tagged phylogenies at year-intervals, imputing the unobserved ancestral antigenic locations for branches that intersect those time points, and summarizing the drift front distance (in AU) from the most recent common ancestor (MRCA) and the mean pairwise distance between locations (in AU) for each lineage at that slice time. The overall antigenic drift rate was calculated by taking the total imputed drift front distance from the lineage MRCA divided by the total time elapsed, measured in AU per year. 95% high posterior density (HPD) estimates were used to measure the uncertainty in these inferences from the posterior sample of trees.

## Other analyses

H1 and H3 influenza A hemagglutinin (HA1) sequences representing the swine reference strains as well as viruses from swine populations in all available geographic regions were compiled from the NIAID Influenza Research Database (IRD) through the web site at http://www.fludb.org on 22/9/2014 (*Squires et al., 2012*) and combined with then unpublished sequences from European swine influenza viruses generated as part of the ESNIP3 project (*Watson et al., 2015*). We added HA1 human seasonal H1 and H3 influenza virus sequences from IRD that represented the entire time period of the study and all human vaccine strain selections.

Nucleotide alignments of the HA1 domain for both subtypes were generated using default settings in MAFFT with subsequent manual correction. For each alignment the sequences were curated for poorly curated data by inferring maximum likelihood phylogenetic trees in IQ-Tree (*Nguyen et al., 2015*) having performed the model test procedure to select the best-fit model. The provisional ML trees were then analyzed using Path-O-Gen and outlier sequences in the root-to-tip regression plots were identified and removed. ML phylogenetic trees were then inferred again using IQ-TREE as above and ultrafast bootstrap analyses performed (*Minh et al., 2013*).

## Acknowledgements

We acknowledge the co-author contributions of members of the ESNIP3 consortium as follows: Scott Reid (AHPA), Montserrat Agüero Garcia, Laboratorio Central de Veterinaria, Spain; Timm Harder, Friedrich-Loeffler-Institut, Germany; Emanuela Foni, Instituto Zooprofilattico Sperimentale delle Venezie, Italy; Iwona Markowska-Daniel, National Veterinary Research Institute, Poland.

## Additional information

### Group author details

ESNIP3 consortium

Scott Reid: Animal and Plant Health Agency, Weybridge, United Kingdom; Montserrat Auero Garcia: Laboratorio Central de Veterinaria, Madrid, Spain; Timm Harder: Friedrich-Loeffler-Institut, Greifswald, Germany; Emanuela Foni: Zooprofilattico Institute of Lombardy and Emilia Romagna, Parma, Italy; Iwona Markowska-Daniel: Department of Swine Diseases, National Veterinary Research Institute, Pulawy, Poland

### Funding

| Funder | Grant reference number | Author |
| --- | --- | --- |
| United States Department of Agriculture-ARS | SCA agreement number 58-3625-2-103F | Nicola S Lewis |

| European Union Seventh Framework Programme | 259949 | Nicola S Lewis<br>Paul Kellam<br>Gaelle Simon<br>Richard J Webby<br>Kristien Van Reeth<br>Sharon M Brookes<br>Ian H Brown<br>Lars Larsen<br>Simon J Watson<br>Amy L Vincent |
|---|---|---|
| Medical Research Council Fellowship | MR/K021885/1 | Judith M Fonville |
| United States Department of Agriculture-ARS | SCA agreement number 58-3625-4-070 | Tavis K Anderson |
| University Research Fellowship from the Royal Society | | Colin A Russell |
| DEFRA and the UK devolved governments | SV34041 | Sharon M Brookes<br>Ian H Brown |

The funders had no role in study design, data collection and interpretation, or the decision to submit the work for publication. Mention of trade names or commercial products in this article is solely for the purpose of providing specific information and does not imply recommendation or endorsement by the U.S. Department of Agriculture.

## Author contributions
NSL, ALV, Conception and design, Acquisition of data, Analysis and interpretation of data, Drafting or revising the article, Contributed unpublished essential data or reagents; CAR, TKA, Conception and design, Acquisition of data, Analysis and interpretation of data, Drafting or revising the article; PLa, DFB, JMF, Analysis and interpretation of data, Drafting or revising the article; KB, PK, Analysis and interpretation of data, Drafting or revising the article, Contributed unpublished essential data or reagents; FB, GD, PLe, Analysis and interpretation of data; RAMF, Conception and design, Analysis and interpretation of data, Drafting or revising the article, Contributed unpublished essential data or reagents; BFK, SJW, Acquisition of data, Analysis and interpretation of data, Drafting or revising the article, Contributed unpublished essential data or reagents; TN, BN, NT, RJW, LL, Acquisition of data, Contributed unpublished essential data or reagents; JSMP, TS, GS, KVR, SMB, IHB, Acquisition of data, Drafting or revising the article, Contributed unpublished essential data or reagents; ES, Conception and design, Analysis and interpretation of data

## Author ORCIDs
Nicola S Lewis, http://orcid.org/0000-0001-9496-4756
Colin A Russell, http://orcid.org/0000-0002-2113-162X
Tavis K Anderson, http://orcid.org/0000-0002-3138-5535
David F Burke, http://orcid.org/0000-0001-8830-3951

## Ethics
Animal experimentation: For sera raised in the U.S., pigs were cared for in compliance with the Institutional Animal Care and Use Committee of the National Animal Disease Centre and for sera raised in the EU, all animal work was done in accordance with the local rules and procedures with ethical permissions granted following the codes of practice for performing scientific studies using animals.

## Additional files

### Supplementary files
• Supplementary file 1. Dimension testing results for antigenic maps characterising the evolution of H1 and H3 influenza viruses.

• Supplementary file 2. Mean pairwise distances between swine influenza lineage groups/strains and between human currently circulating strains (AU) and associated 95% credible interval.

• Supplementary file 3. Overall drift rate in antigenic units per year for H1 and H3 swine influenza virus lineages and 95% credible interval (HPD).

• Supplementary file 4. Summary of previously reported rates of antigenic drift of influenza A viruses.

### Major datasets

The following dataset was generated:

| Author(s) | Year | Dataset title | Dataset URL | Database, license, and accessibility information |
|---|---|---|---|---|
| Lewis NS, Russell CA, Langat P, Anderson TK, Berger K, Bielejec F, Burke DF, Dudas G, Fonville JM, Fouchier RAM, Kellam P, Koel BF, Lemey P, Nguyen T, Nuansrichy B, Peiris JSM, Saito T, Simon G, Skepner E, Takemae N, ESNIP3 consortium, Webby R, Van Reeth K, Brookes SM, Larsen L, Watson SJ, Brown IH, Vincent AL | 2016 | Data from: The global antigenic diversity of swine influenza A viruses | http://dx.doi.org/10.5061/dryad.5ff80 | Available at Dryad Digital Repository under a CC0 Public Domain Dedication |

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
