## [Decision Letter]

Thank you for submitting your work entitled "Quantifying the global antigenic diversity of swine influenza A viruses" for consideration by *eLife*. Your article has been reviewed by three peer reviewers, including Vijaykrishna Dhanasekaran, and the evaluation has been overseen by Mark Jit as the Reviewing Editor and Prabhat Jha as the Senior Editor.

The reviewers have discussed the reviews with one another and the Reviewing Editor has drafted this decision based on their comments. All of them agreed that this is a well written, comprehensive study of the antigenic diversity of swine influenza A viruses. The effort in assembling virus samples from several labs in Europe, USA, and Asian countries to carry out HI assays is a significant effort, and rectifies the lack of antigenic data from swine in recent years. The analysis adds important information to inform vaccine design and risk assessment of viruses that may present human threats. They also agreed that the antigenic analysis is the strength and novelty of this study, and could be expanded on significantly.

Essential revisions:

The reviewers and editors had two major concerns both of which need to be addressed. We would like to give you the opportunity to submit a revised version addressing the two major concerns within two months.

1) The antigenic maps presented in Figure 2 and supporting information are difficult to understand and interpret. Furthermore, the description that the antigenic diversity is greater or smaller is largely descriptive throughout the manuscript. To alleviate both problems, we suggest that you use the methods in your previous paper (Bedford et al. 2014), which integrates both genetic and antigenic data.

2) The reviewers all agreed that the greatest weakness of this manuscript is the phylogenetic analysis. The dataset used to generate Figure 1 includes several erroneously generated sequences. However, if you decide to use the Bedford et al. (2014) method as they suggested, the phylogenetic analysis could be dropped entirely, as it simultaneously characterises antigenic and genetic evolution and provides a method to visualize both simultaneously.

We have listed the issues identified by the reviewers with the phylogenetic analysis below. Even if you decide to address them rather than to drop this section entirely, we suggest that the large amount of text describing the phylogenetic analysis in the Results section is removed and replaced with further analysis of antigenic evolution.

A) The analysis has not inferred any ML trees. Relying just on BEAST trees is going to produce errors Bayesian time-scaled trees are very good at obscuring contaminants and sequencing errors. Most glaringly, the claim that there are 2 main lineages of classical H1N1 viruses arose because ML trees were not made. Colombia/0401, Guangdong/L3, and StHyacinthe/148 – these are all clearly sequencing errors if you make an ML tree, and not a second classical lineage that has managed to persist mostly undetected in swine for five decades, as the authors claim. The maximum-likelihood phylogeny in conjunctions with root-to-tip regression (using software such as Path-o-Gen) can be used easily to identify and remove sequences prior to BEAST analysis.

B) The legend says the branch colors are supposed to represent cross-species introductions, but there are plenty of examples where independent human-to-swine introductions (described in previously published literature) are colored the same (the green clade on the δ tree is a particularly glaring example of this, with at least 5 separate human-to-swine introductions all shaded green as if a single introduction – if the authors used more human background data this would be readily apparent). Although the text states that there are 36 separate human-to-swine transmission events of human H3N2 seasonal viruses, these are not labeled on the tree and there don't appear to be nearly this many, at least that currently circulate.

C) The presentation of the phylogenetic trees is perplexing. Why are there node labels for the age of nodes, when this can be determined from the x-axis already, but no node labels for node support (posterior probabilities), the key indication of clades and topological robustness? The shaded circles are inane – you can tell the length of the branch just from looking at the tree.

D) The data set is not well curated. There are avian-origin viruses in the tree (e.g., Saskatchewan/18789) that are presented as part of the avian-like Eurasian lineage, which incorrectly dates the tMRCA for this lineage all the day back to 1964 (when it should be the late 1970s).

[Editors' note: further revisions were requested prior to acceptance, as described below.]

Thank you for resubmitting your work entitled "The global antigenic diversity of swine influenza A viruses" for further consideration at *eLife*. Your revised article has been favorably evaluated by Prabhat Jha as the Senior editor, Mark Jit as the Reviewing editor, and three reviewers, one of whom, Vijaykrishna Dhanasekaran, has agreed to reveal his identity.

The manuscript has been improved but there are some remaining issues that need to be addressed before acceptance, as outlined below.

The reviewers agreed that the revised manuscript is a commendable and substantial improvement over the first submission. In particular they appreciated the work in correcting the phylogenies, using the BMDS approach to quantify rates of antigenic drift and new figures which better reflect the genetic and antigenic variation among the different lineages.

Their chief remaining concern is with the 'risk profiles for the global movement of swine and the potential for swine influenza-derived infections in humans.' The manuscript uses antigenic distances between viruses circulating in pig populations in different countries and in humans as a way to predict the likelihood of (a) viruses from one pig population invading another; or (b) transmitting successfully to humans, either as an outbreak or pandemic. The issue with (a) is that competition dynamics between strains are poorly understood in swine. There are repeated introductions of similar HAs and NAs into the same swine population, often with co-circulation. There is anecdotal evidence that the lack of onward transmission of the pandemic H1 in US and other swine populations is related to strain competition with not too distantly related classical H1s. But the idea that the probability of invasion is positively correlated with antigenic distance is an oversimplification and not based on any empirical evidence. It also fails to take into account reassortment, and even if the HA is outcompeted other segments can persist (as has been the case with the pandemic virus). For (b), it is clear that the restrictions on an animal virus successfully transmitting to humans have relatively little to do with antigenic properties. Antigenic distances are very likely to predict the age-specific attack rate of a pandemic virus, skewing the burden towards younger age groups. But the notion that antigenic distance is a good predictor of the likelihood of the pandemic occurring in the first place is not supported by any evidence. The paper relies on anecdotal evidence from H3N2v that it has not caused a pandemic due to existing immunity in adults. But this did not stop the pandemic of 1977. Your manuscript gets credit for stating up front that antigenic distance is not likely to be a key factor in pandemic emergence, but it then contains maps that likely misrepresent pandemic risk.

Hence we feel that (i) Figure 3 should be removed because it is based on the unjustified premise that antigenic distance is predictive of viral invasion or pandemic emergence, and (ii) the discussion of risk assessment should be qualified with the caveats above.

---

## [Author Response]

Essential revisions: The reviewers and editors had two major concerns both of which need to be addressed. We would like to give you the opportunity to submit a revised version addressing the two major concerns within two months.

*1) The antigenic maps presented in Figure 2 and supporting information are difficult to understand and interpret. Furthermore, the description that the antigenic diversity is greater or smaller is largely descriptive throughout the manuscript. To alleviate both problems, we suggest that you use the methods in your previous paper (Bedford et al. 2014), which integrates both genetic and antigenic data.* We have now re-analyzed all the antigenic data from the original manuscript using the BMDS methodology developed in Bedford et al. 2014. These new analyses allowed for the direct integration of the virus antigenic and genetic data and have allowed for the creation of new figures that improve the interpretability of the antigenic maps.

We have substantially re-written the manuscript to reflect these new analyses and replaced several of the figures from the original manuscript with new figures based on the BMDS analyses.

We have also quantified antigenic variation by genetic lineage, by geographic location of sample collection (Figure 1), and the rate of antigenic drift of each lineage since time of introduction into swine (Figure 2).

*2) The reviewers all agreed that the greatest weakness of this manuscript is the phylogenetic analysis. The dataset used to generate Figure 1 includes several erroneously generated sequences. However, if you decide to use the Bedford et al. (2014) method as they suggested, the phylogenetic analysis could be dropped entirely, as it simultaneously characterises antigenic and genetic evolution and provides a method to visualize both simultaneously. We have listed the issues identified by the reviewers with the phylogenetic analysis below. Even if you decide to address them rather than to drop this section entirely, we suggest that the large amount of text describing the phylogenetic analysis in the Results section is removed and replaced with further analysis of antigenic evolution.* We have carefully reviewed all genetic sequence data used in our analyses. Our original analyses had inadvertently included some likely mis-labeled or mis-dated sequences that should have been excluded from our analyses. We analyzed all of the genetic data included in the revised manuscript using Path-o-Gen and removed all outlier viruses. The plots below for H1 and H3 viruses show the root-to-tip distances for the final genetic dataset – a subset of which was used in the BMDS analyses.

In addition to the BMDS analyses described above we have elected to include the ML phylogenetic trees as supplemental material. As the reviewers well know, swine influenza surveillance has strong geographic biases and antigenic data is less plentiful than genetic sequence data. By analyzing genetic sequence data for which we lack complimentary antigenic data we are able to infer important gaps in the global antigenic data that likely obscure the true global antigenic diversity of swine influenza viruses (see subsection “Additional antigenic diversity can be inferred from genetic sequence data”). We feel that it is important to include such phylogenetic analyses of viruses for which we do not have antigenic data to further inform the reader as to the potential, but as yet uncharacterized, antigenic diversity. These genetic analyses are not the most novel aspects of our study – thus we decided to include the re-inferred ML trees as supplemental material.

*A) The analysis has not inferred any ML trees. Relying just on BEAST trees is going to produce errors Bayesian time-scaled trees are very good at obscuring contaminants and sequencing errors. Most glaringly, the claim that there are 2 main lineages of classical H1N1 viruses arose because ML trees were not made. Colombia/0401, Guangdong/L3, and StHyacinthe/148 – these are all clearly sequencing errors if you make an ML tree, and not a second classical lineage that has managed to persist mostly undetected in swine for five decades, as the authors claim. The maximum-likelihood phylogeny in conjunctions with root-to-tip regression (using software such as Path-O-Gen) can be used easily to identify and remove sequences prior to BEAST analysis.* We have followed this advice and in the revised manuscript we have included ML trees as supplemental material. These trees were constructed after carefully curating the sequence data to ensure the removal of erroneous sequences.

*B) The legend says the branch colors are supposed to represent cross-species introductions, but there are plenty of examples where independent human-to-swine introductions (described in previously published literature) are colored the same (the green clade on the δ tree is a particularly glaring example of this, with at least 5 separate human-to-swine introductions all shaded green as if a single introduction – if the authors used more human background data this would be readily apparent). Although the text states that there are 36 separate human-to-swine transmission events of human H3N2 seasonal viruses, these are not labeled on the tree and there don't appear to be nearly this many, at least that currently circulate.* We have removed these trees and the associated text. All phylogenetic inferences are now based on the two maximum likelihood trees included as supplemental material and on the incorporation of the genetic data into the BMDS methodology.

*C) The presentation of the phylogenetic trees is perplexing. Why are there node labels for the age of nodes, when this can be determined from the x-axis already, but no node labels for node support (posterior probabilities), the key indication of clades and topological robustness? The shaded circles are inane – you can tell the length of the branch just from looking at the tree.* We have removed these figures and replaced them with two ML phylogenetic trees (Figure 1—figure supplement 3). The coloring and marking schemes of the new ML trees is simple and easier to interpret than the figures included in the original manuscript.

D) The data set is not well curated. There are avian-origin viruses in the tree (e.g., Saskatchewan/18789) that are presented as part of the avian-like Eurasian lineage, which incorrectly dates the tMRCA for this lineage all the day back to 1964 (when it should be the late 1970s).

These sequences were identified by Path-o-Gen during our review of the data and removed from our analyses.

[Editors' note: further revisions were requested prior to acceptance, as described below.]

*The manuscript has been improved but there are some remaining issues that need to be addressed before acceptance, as outlined below. The reviewers agreed that the revised manuscript is a commendable and substantial improvement over the first submission. In particular they appreciated the work in correcting the phylogenies, using the BMDS approach to quantify rates of antigenic drift and new figures which better reflect the genetic and antigenic variation among the different lineages. Their chief remaining concern is with the 'risk profiles for the global movement of swine and the potential for swine influenza-derived infections in humans.' The manuscript uses antigenic distances between viruses circulating in pig populations in different countries and in humans as a way to predict the likelihood of (a) viruses from one pig population invading another; or (b) transmitting successfully to humans, either as an outbreak or pandemic. The issue with (a) is that competition dynamics between strains are poorly understood in swine. There are repeated introductions of similar HAs and NAs into the same swine population, often with co-circulation. There is anecdotal evidence that the lack of onward transmission of the pandemic H1 in US and other swine populations is related to strain competition with not too distantly related classical H1s. But the idea that the probability of invasion is positively correlated with antigenic distance is an oversimplification and not based on any empirical evidence. It also fails to take into account reassortment, and even if the HA is outcompeted other segments can persist (as has been the case with the pandemic virus). For (b), it is clear that the restrictions on an animal virus successfully transmitting to humans have relatively little to do with antigenic properties. Antigenic distances are very likely to predict the age-specific attack rate of a pandemic virus, skewing the burden towards younger age groups. But the notion that antigenic distance is a good predictor of the likelihood of the pandemic occurring in the first place is not supported by any evidence. The paper relies on anecdotal evidence from H3N2v that it has not caused a pandemic due to existing immunity in adults. But this did not stop the pandemic of 1977. Your manuscript gets credit for stating up front that antigenic distance is not likely to be a key factor in pandemic emergence, but it then contains maps that likely misrepresent pandemic risk. Hence we feel that (i) Figure 3 should be removed because it is based on the unjustified premise that antigenic distance is predictive of viral invasion or pandemic emergence, and (ii) the discussion of risk assessment should be qualified with the caveats above.*

We thank the reviewers for highlighting these important issues for influenza virus research and for suggesting a more nuanced approach.

We acknowledge that there is currently an incomplete understanding of the factors involved the establishment of swine viruses from one pig population into another and for viruses emerging from pigs to infect and potentially transmit among humans. However, both HA antigenic diversity and prior population immunity within pigs and humans are key factors when assessing these risks and are specifically included in the US Centers for Disease Control’s Influenza Risk Assessment Tool (Trock et al., Avian Diseases 2012). However, we acknowledge that solely discussing HA in reference to Figure 3 could be misleading. Thus we have removed Figure 3, not because we think that phenotypic differences are unimportant but so as not to oversimplify the discussion of the factors that shape risk.

We have also added the requested caveats to the Discussion to make clear that antigenic variation is only one of several traits that contribute to the risk profile of swine influenza viruses and factors unrelated to antigenic distance also influence the ability of non-HA gene segments from one population being maintained in another through genetic reassortment with endemic strains. Specifically, we added following text to the Discussion:

“Neutralizing immunity to influenza viruses in both humans and swine is largely directed towards the influenza HA and the antigenic dissimilarity among influenza HAs within and between these two hosts is undoubtedly important for influenza epidemiology. […] There is also a need to better understand the competition dynamics of different influenza virus variants and subtypes in both human and swine populations and how these dynamics affect the potential for invasion by novel viruses.”

We have also made minor changes to other portions of the text to ensure that we have not overstated the importance of antigenicity in assessing risk.